# Efficacy of a Cosmetic Treatment in Decreasing the Mild-to-Moderate Atopic Dermatitis in Babies, Children, and Adults: A Pilot Study

Vincenzo Nobile [1,*], Valentina Zanoletti [1], Marta Pisati [2] and Enza Cestone [2]

[1] R&D Department, Complife Italia Srl, 27028 San Martino Siccomario, Italy
[2] Clinical Testing Department, Complife Italia Srl, 27028 San Martino Siccomario, Italy
* Correspondence: vincenzo.nobile@complifegroup.com; Tel.: +39-0382-25504

**Abstract:** Atopic dermatitis (AD) is a chronic inflammatory and pruritic skin disease with a worldwide progressive increase in its incidence. In this clinical study, we studied the effect of a cosmetic treatment composed of a cleanser, and a body and face cream, on subjects (babies, children, and adults) suffering from mild-to-moderate AD. The product effect on AD clinical signs was investigated by SCORing Atopic Dermatitis (SCORAD) index, subjective evaluation, skin erythema index, and transepidermal water loss (TEWL) measurements. The products were shown to be effective in improving the AD scoring by SCORAD in all the groups, and a trend towards the decrease of the erythema index and the TEWL in the adult population. An improvement in itching sensation, skin redness, and skin dryness scoring was also reported by the subjects. Results from this study demonstrate the efficacy of the tested products in decreasing the overall AD severity through 28 days of treatment. Overall, the first results occurred within 14 days of treatment.

**Keywords:** atopic dermatitis; cosmetic treatment; cleanser; body cream; face cream





## 1. Introduction

Atopic dermatitis (AD) is a chronic inflammatory and pruritic skin disease arising during the first months of life or at maturity [1]. The clinical signs of AD consist of eczema-like eruptions, including erythema, papules (swelling), exudative lesions (oozing/crusting), scratch marks, skin thickening (lichenification), and dryness [2–4]. The worldwide incidence of AD shows a progressive increment [5] and now it is estimated that it affects 20–25% of children [6–9] and 2–7% of adults with regional differences [2,10]. Given its impact on sleep, emotional and mental health, physical activity, and social functioning, the quality of life of AD patients is impaired [11,12].

AD is a multifactorial etiopathogenesis skin disease. Dysbiosis of the skin microbiota, genetic and environmental factors, altered immune response, and epidermal barrier disruption are the main factors involved in the pathogenesis of AD [4]. However continuous studies on the pathogenetic pathway of atopic dermatitis keep shedding light on the pathophysiology of the disease and are aimed to identify new key molecules that should have a practical impact in the therapeutic field.

The overall AD severity is assessed by both objective and subjective symptoms [13] using the SCORing Atopic Dermatitis (SCORAD) index [14]. According to the SCORAD index, mild AD corresponds to SCORAD levels below 15, moderate AD to SCORAD levels in the range of 15–40, while severe AD to SCORAD levels above 40 [15].

The guidelines for AD treatment are based on the severity of the AD symptoms. The restoration of the epidermal barrier function by emollients is generally the target in the prevention and treatment of mild-to-moderate AD. Emollients are also indicated in the first month of life when the first presentation of AD known as "cradle cap" appears [16]. The

application of emollients is recommended on humid skin using the so-called "soak-and-seal" technique [17]. The emollient therapy is cost-effective, on non-inflamed skin, because of their ability to prevent the development of flares and the need for a pharmaceutical approach using topical corticosteroids or tacrolimus ointments. The European Task Force on Atopic Dermatitis (ETFAD) recommends using daily at least 30 g of emollients, which should be preferentially applied in a 'soak-and-seal' technique [18].

This study investigated the efficacy of a treatment based on daily use of three cosmetic products in three separate groups of subjects divided by age range (babies, children, and adults) and suffering from mild-to-moderate AD. The cosmetic products used in this randomized placebo-controlled clinical trial were a liquid emollient cleanser, an emollient cream for the body, and an emollient facial cream. All the cosmetic products contained in their formula INDUFENCE® (Silab, Saint-Viance, France), a raw material derived from Alisma Plantago-aquatica, claiming the following activities; stimulation of both the mechanical and the immune biological barrier of the skin by a probiotic-like effect, activation and optimization of the skin natural defenses, and decrease in the inflammation induced by the microbial (*S. aureus*) aggression; vegetal ceramides from rice, which reinforce the barrier function of the skin supporting both the skin immediate hydration and the skin long-lasting moisturization; and a mixture of gluco-oligosaccharides and inulin which regulates the balance of skin microbiota (BIOLIN/P, GOVA, Antwerp, Belgium).

## 2. Materials and Methods

### 2.1. Human Volunteers and Study Design

A randomized, double blind, placebo-controlled, multicentric clinical trial was carried out in accordance with the World Medical Association's (WMA) Helsinki Declaration and its amendments on three groups of twenty ($n = 20$) subjects of different age showing mild-to-moderate SCORAD levels of AD (15 < SCORAD < 40). A list of both the inclusion and exclusion criteria can be found in Supplementary Table S1. Subjects agreed to stop topical or systemic treatments at least 2 weeks before the start of the study and not to use other topical or systemic substances for the treatment of dermatitis during the study period. The groups' composition was as follows:

- Group 1 (G1): Twenty ($n = 10$ active and $n = 10$ placebo) male and female newborns/infants/toddlers (6–36 months), enrolled by a pediatrician and a dermatologist;
- Group 2 (G2): Twenty ($n = 10$ active and $n = 10$ placebo) male and female children (3–14 years), enrolled by a dermatologist;
- Group 3 (G3): Twenty ($n = 10$ active and $n = 10$ placebo) male and female adults (18+ years) enrolled by a dermatologist.

The clinical trial on G1 and G3 was carried at Complife Italia Srl facilities in Pavia (San Martino Siccomario, PV, Italy); while the study on G2 was carried out at Nutratech (a Complife Italia company) facility in Rende (CS, Italy).

Both the study protocol and the informed consent forms were reviewed and approved (ref. no. 2021/17 by 23 December 2021) by the "Comitato Etico Indipendente per le Indagini Cliniche Non Farmacologiche" (Società Scientifica Italiana per le Indagini Cliniche Non Farmacologiche. Genova, Italy).

### 2.2. Intervention

The tested treatment were three commercially available cosmetic products (Novatopia emollient milk bath, Novatopia emollient cream, Novatopia emollient ato-balance face cream supplied by Rontis Hellas S.A., Athens, Greece (now Dreavia AG, Zug, Switzerland). All the products contained natural peptides derived from Alisma Plantago-aquatica (INDUFENCE® (Silab, Saint-Viance, France), vegetal ceramides from rice, and a mixture of gluco-oligosaccharides and inulin (BIOLIN/P, GOVA, Antwerp, Belgium). The active and placebo products composition can be found in Supplementary Table S2. The active and placebo products way of use was as follows:

- Novatopia emollient milk bath: Apply a quantity of novatopia emollient milk bath to wet skin and scalp and gently clean. Rinse off thoroughly and gently pat the skin. For optimal skin hydration, use in combination with novatopia emollient cream. This product was used to clean the body, the face, and the scalp/hair;
- Novatopia emollient cream: Apply the novatopia emollient cream 2–3 times a day to dry skin and massage gently. For optimal skin hydration, use in combination with novatopia emollient milk bath. This product was applied all over the body;
- Novatopia emollient ato-balance face cream: Apply twice daily to clean and damp skin, avoiding the eye area. For optimal skin hydration, apply after cleaning with novatopia emollient milk bath. This product was applied on the face.

### 2.3. Randomization and Masking

Enrolled subjects from each group were randomly assigned according to a restricted randomization list generated by an appropriate statistic algorithm ("Wey's urn") to receive the blinded active or placebo treatment. The randomization list was generated using PASS 11 (version 11.0.8; PASS, LLC. Kaysville, UT, USA) statistical software running on Windows Server 2008 R2 Standard SP1 64-bit edition (Microsoft, Redmond, WA, USA). For each subject participating in the study an envelope was prepared containing the information on the tested products. Both the randomization list and the subjects' envelopes were stored by the in-site Study Director under appropriate safety conditions in a place that was not accessible neither to volunteers nor to the experimenter.

### 2.4. Methods

The parameters reported here below were assessed after 15–20 min of acclimatization under controlled ambient conditions (T = 18–26 °C and RH = 40–60%). The study flow and schedule of assessments chart is reported in Table 1.

**Table 1.** Study flow and schedule of assessment chart.

| Study Phases | Initial Visit-Start of the Study (T0) | | | Intermediate Visit (T14) | | | Final Visit (T28) | | |
|---|---|---|---|---|---|---|---|---|---|
| | G1 | G2 | G3 | G1 | G2 | G3 | G1 | G2 | G3 |
| Signed Informed consent (for minors filled-in by parent or legal guardian) | X | X | X | | | | | | |
| Subject eligibility * | X | X | X | X | X | X | X | X | X |
| Products distribution | X | X | X | | | | | | |
| SCORAD evaluation | X | X | X | X | X | X | X | X | X |
| Itching evaluation on VAS scale | | X | X | | X | X | | X | X |
| Weekly diary distribution | X | X | X | | | | | | |
| Weekly diary collection | | | | | | | X | X | X |
| Instrumental evaluation: TEWL (Tran Epidermal Water Loss) | | X | | | X | | | X | |
| Instrumental evaluation: Erythema index | | X | | | X | | | X | |
| Self-assessment questionnaire (for minors filled-in by parent or legal guardian) | | | | | | | X | X | X |
| AE and local tolerance assessment | | | | X | X | X | X | X | X |
| Product collection | | | | | | | X | X | X |

* All evaluations were performed on cleaned skin. The last application of the product was at least 24 h before the visit.

### 2.4.1. SCORAD

The SCORing Atopic Dermatitis index (SCORAD) is a well-established severity-scoring tool [14] for atopic dermatitis (AD). The calculation of SCORAD is based on the extent of lesions, on the intensity of lesions (erythema, edema/papulation, oozing/crusts, excoriations, lichenification, dryness), and on subjective symptoms scoring (itching and sleep loss). Itching is defined as moderate if it is present up to 10% of the time and interferes with the ability for daily living. It is defined as severe if it is present most of the time and makes the individual wake up at night. If fissures are present, the score is moderate or severe. This item was evaluated with the parents' collaboration for groups 1 and 2. The itching scoring on the VAS scale taken during the SCORAD evaluation is analyzed separately from the other parameters.

### 2.4.2. Transepidermal Water Loss (TEWL)

The transepidermal water loss (TEWL, perspiratio insensibilis) was measured using a Tewameter® TM 300 (Courage+Khazaka Electronic, Köln, Germany). Tewameter® probe measures indirectly the density gradient of water evaporation over the skin surface using two pairs of sensors (temperature and relative humidity) in an "open chamber" configuration mode. The Fick diffusion law is the basis for the measurement allowing for the calculation of the evaporation rate in $g \cdot h^{-1} \cdot m^{-2}$. Precautions were taken to avoid any turbulence all over the measurement area. Measurements were performed on a skin area showing AD signs on the face and body.

### 2.4.3. Erythema Index

The erythema index was measured with Mexameter® MX 18 (Courage+Khazaka Electronic, Köln, Germany) which specifically measures hemoglobin content (erythema) in the skin. Measurements were performed on a skin area showing AD signs on the face and body.

### 2.4.4. Weekly Diary

The parents/subjects were asked to fill, weekly, a diary to assess the progression and the perception of the severity of the following discomforts: itching, skin redness, and skin dryness. These discomforts were scored on an 11-point visual analog scale (VAS), from 0 = no skin discomfort to 10 = worst skin discomfort possible.

### 2.4.5. Self-Assessment Questionnaire

The parents/subjects were asked to answer the questions of a self-assessment questionnaire about the tolerability, efficacy, and pleasantness of the tested treatment. Each item of the questionnaire was scored on a 4-point scale (completely agree, agree, disagree, and completely disagree).

### 2.4.6. Statistical Analysis

Data were summarized using frequency distributions (number and percentage) for categorical/ordinal variables. For continuous variables, the following values were calculated: mean value, minimum value, maximum value, standard error of the mean (SEM), individual variation/individual percentage variation, and mean variation/mean percentage variation. All the calculations were done using a Microsoft® Excel 365 (vers. 2304; build 16327.20214; Microsoft, Redmond, WA, USA) worksheet running on Microsoft® Windows 11 Professional (vers. 22H2, build SO 22621.1635 Microsoft, Redmond, WA, USA). The results of the self-assessment questionnaire were calculated as a percentage (%) of subjects who were assigned a particular judgment (among those proposed). For each question, the number of subjects related to each judgment was counted → (number of subjects) and then divided by the total number of subjects → % of answers.

Data normality was checked by the Shapiro–Wilk normality test before any statistical procedure. The data distribution of all the parameters was not normal. A Wilcoxon

(intragroup analysis) or Mann–Whitney test (intergroup analysis) was used for statistical analysis. The statistical analysis was carried out using NCSS 10 (version 10.0.7 for Windows; NCSS, Kaysville, UT, USA) running on Windows Server 2008 R2 Standard SP1 64-bit edition (Microsoft, Redmond, WA, USA). A $p < 0.05$ was considered statistically significant. The level of significance was reported as follows: * $p < 0.05$, ** $p < 0.01$, and *** $p < 0.001$.

## 3. Results

The clinical study was carried out from February to June 2022. A total of 60 subjects were successfully randomized. Thirty ($n = 30$) subjects were allocated to each treatment arm (Figure 1).

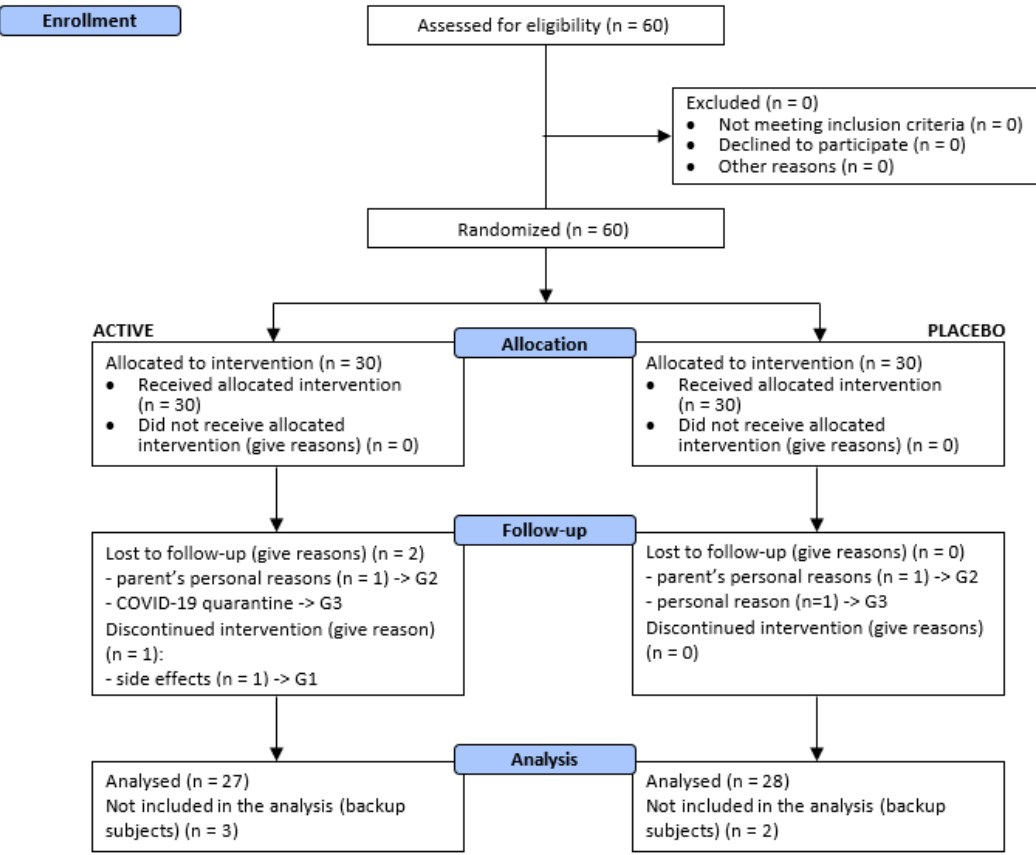

**Figure 1.** Participants flow diagram. G1 newborns/infants/toddlers (6–36 months); G2 children (3–14 years), enrolled by a dermatologist; G3 adults (18+ years).

The demographic features of participants are shown in Table 2. Both treatments were generally well tolerated by all the subjects and no adverse events were reported during the study period. One baby dropped out soon after the beginning of the study; occurred reactions were classified by the pediatrician as likely related to product use and probably due to individual sensitivity to the product itself or to ingredient(s) of the product. Parents of two children decided to withdraw from the study for personal reasons, not related to the treatments. One adult subject was lost to follow-up for personal reasons and one adult subject was lost to follow-up for COVID-19 quarantine.

Fifty-five ($n = 55$) subjects completed the study as per protocol, as follows:

- G1: 19 subjects (active treatment $n = 9$ and placebo $n = 10$);
- G2: 18 subjects (active treatment $n = 9$ and placebo $n = 9$);
- G3: 18 subjects (active treatment $n = 9$ and placebo $n = 9$).

**Table 2.** Baseline and demographic characteristics. G1 newborns/infants/toddlers (6–36 months); G2 children (3–14 years); G3 adults (18+ years). n.a. not applicable or not measured. Data are mean ± SE.

|  | **Active** | | | **Placebo** | | |
|---|---|---|---|---|---|---|
|  | **G1** | **G2** | **G3** | **G1** | **G2** | **G3** |
| Sex |  |  |  |  |  |  |
| Male (*n*) | 6 | 4 | 2 | 7 | 4 | 1 |
| Female (*n*) | 3 | 5 | 7 | 3 | 5 | 8 |
| SCORAD index | 19.2 ± 0.8 | 23.2 ± 0.2 | 29.0 ± 3.2 | 17.5 ± 0.6 | 24.8 ± 3.0 | 27.7 ± 2.2 |
| Overall SCORAD index | 23.8 ± 1.7 | n.a. | n.a. | 23.4 ± 1.4 | n.a. | n.a. |
| Itching score | n.a. | 5.7 ± 0.6 | 3.8 ± 0.6 | n.a. | 5.3 ± 0.6 | 4.2 ± 0.7 |
| Overall itching score | 4.8 ± 0.5 | n.a. | n.a. | 4.7 ± 0.5 | n.a. | n.a. |
| TEWL | n.a. | n.a. | 31.6 ± 6.1 | n.a. | n.a. | 33.3 ± 5.7 |
| Erythema index | n.a. | n.a. | 407.9 ± 37.8 | n.a. | n.a. | 379.6 ± 25.0 |

**SCORAD**. A progressive and significant reduction of the SCORAD index was recorded in all groups receiving the active treatment, as well as in the G1 (only after 14 days of treatment), G2, and G3 groups receiving the placebo treatment (Figure 2). The SCORAD index variation in the active treatment arms was generally higher compared to the placebo treatment arms even if a statistically significant difference vs. placebo was only seen in the G1 and in the overall group.

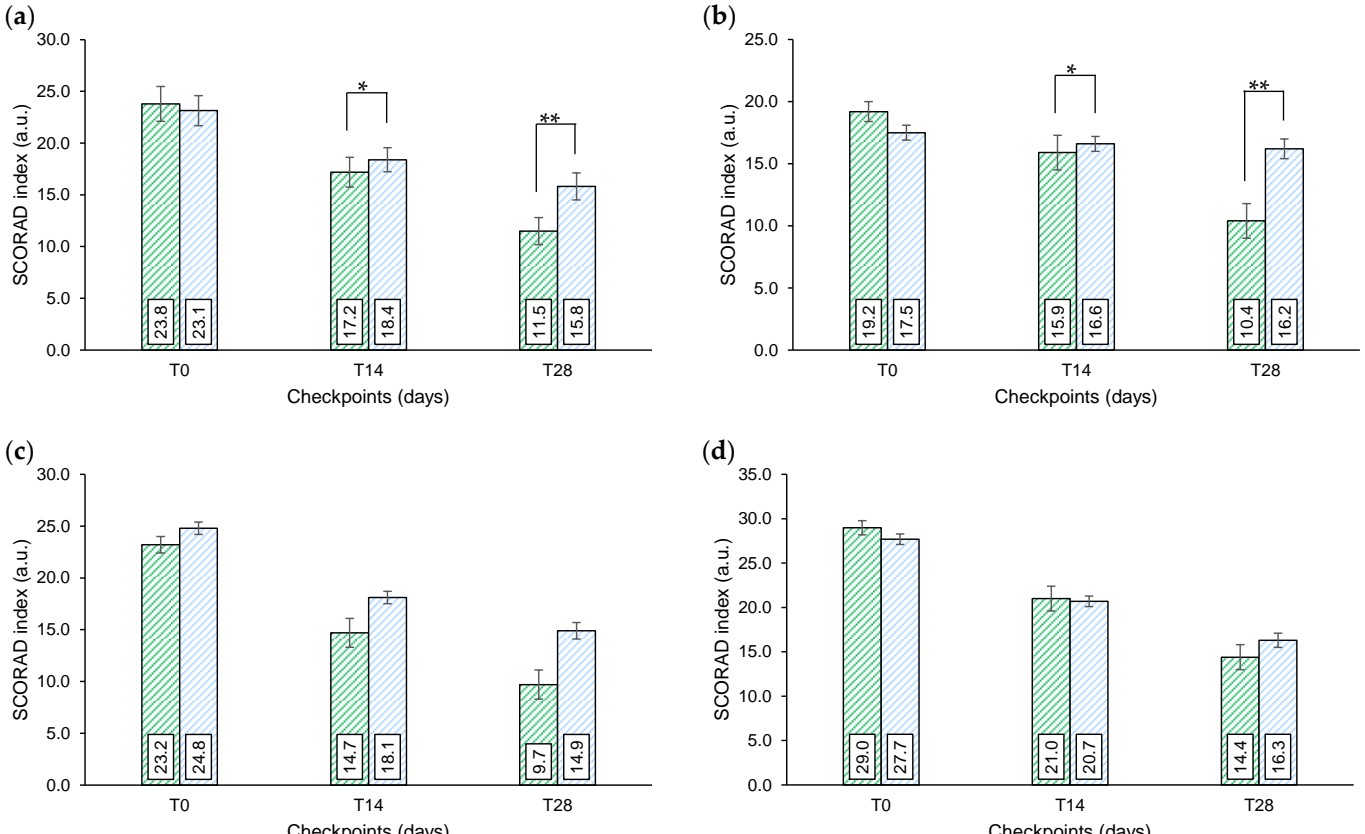

**Figure 2.** SCORAD index. (**a**) Overall (G1 + G2 + G3) SCORAD index. (**b**) G1 SCORAD index. (**c**) G2 SCORAD index. (**d**) G3 SCORAD index. G1 newborns/infants/toddlers (6–36 months); G2 children (3–14 years); G3 adults (18+ years). Above the bars is reported the intergroup statistical analysis as follows: * $p < 0.05$, ** $p < 0.01$.

When analyzed separately, the overall (G2 + G3) itching scoring decrease (% variation) after 28 days of product use was higher in the active treatment arm when compared to the placebo treatment arm (Table 3). The itching score decrease (vs. baseline) was statistically significant in all the treatment groups (active and placebo).

**Table 3.** Itching scoring. G1 newborns/infants/toddlers (6–36 months); G2 children (3–14 years); G3 adults (18+ years). The itching sensation was scored on a 11-point visual analog scale (where 0 = no itching and 10 = worst imaginable itching sensation) * $p < 0.05$ (intergroup statistical analysis on percentage variation).

| Itching Scoring | Active | | | Placebo | | |
| --- | --- | --- | --- | --- | --- | --- |
| | T0 | T14 | T28 | T0 | T14 | T28 |
| Overall | $4.8 \pm 0.5$ | $2.8 \pm 0.5$ | $1.6 \pm 0.4$ * | $4.7 \pm 0.5$ | $3.4 \pm 0.4$ | $2.8 \pm 0.4$ |
| G2 | $5.7 \pm 0.8$ | $3.8 \pm 0.7$ | $2.2 \pm 0.7$ | $5.3 \pm 0.6$ | $4.4 \pm 0.4$ | $3.8 \pm 0.4$ |
| G3 | $3.8 \pm 0.6$ | $1.8 \pm 0.6$ | $1.1 \pm 0.5$ | $4.2 \pm 0.7$ | $2.4 \pm 0.5$ | $1.8 \pm 0.6$ |

**Weekly diary output**. During the study duration subjects/parents were asked to score on an 11-point VAS scale the following discomforts: itching sensation, skin redness, and skin dryness. Due to the reduced number of subjects for group, and the subjectivity of the response, data referred by parents or recorded by the subjects, were pooled by symptom and for each treatment. Results were analyzed by comparing results achieved at the end of week 2–3 and 4 with respect to week 1 (Figure 3).

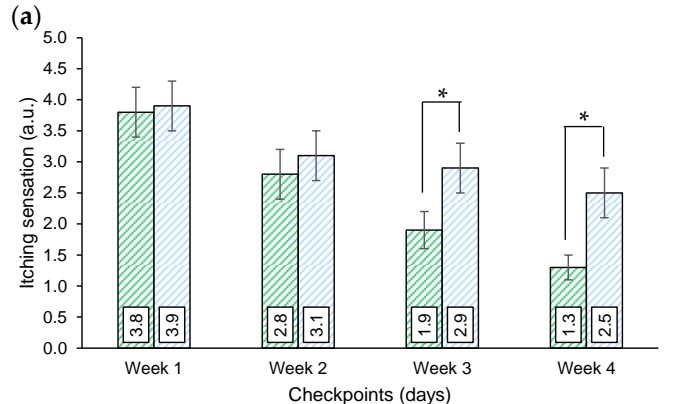
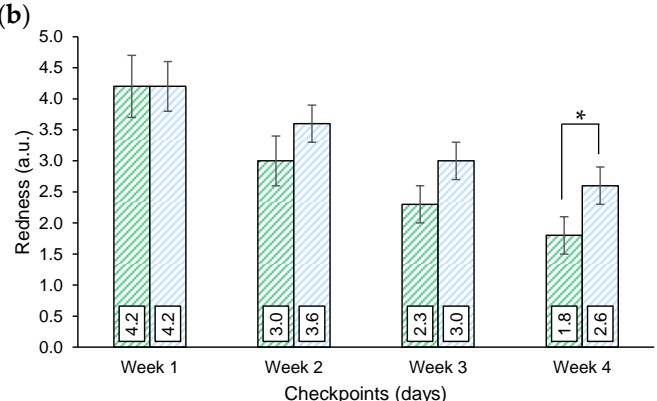

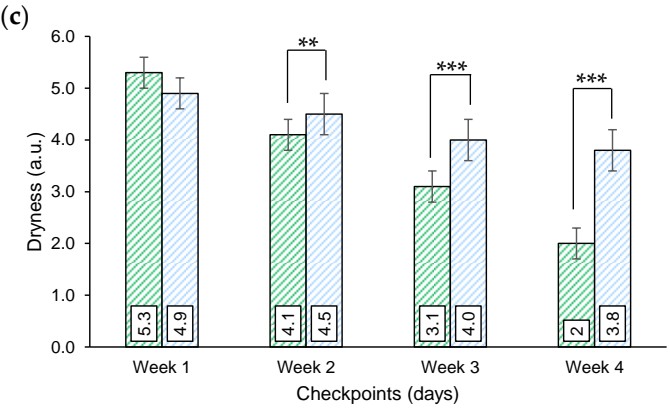

**Figure 3.** Weekly diary output. (**a**) Itching sensation scoring. (**b**) Skin redness scoring. (**c**) Skin dryness scoring. The parameters were scored on an 11-point visual analog scale (where 0 = absence of the condition/sensation and 10 = worst imaginable condition/sensation). Above the bars is reported the intergroup statistical analysis as follows: * $p < 0.05$, ** $p < 0.01$, *** $p < 0.001$.

Aggregated VAS scores for itching indicate that a progressive and significant reduction, with respect to week 1, was achieved by both the active and the placebo treatments throughout the study. The reduction, however, was higher in the active treatment arm and resulted in a significative intergroup difference after 3 and 4 weeks of treatments.

A similar trend was achieved by plotting aggregated VAS scores for skin redness and skin dryness. Both the active and the placebo treatments resulted in a progressive and significative reduction of the discomfort with respect to week 1, the reduction was higher in the active group and resulted in a significative intergroup difference after 4 weeks of treatments for skin redness and at weeks 2, 3, and 4 for skin dryness.

The overall judgment of the active treatment (G1 + G2 + G3) was higher when compared to the placebo arm (Figure 4).

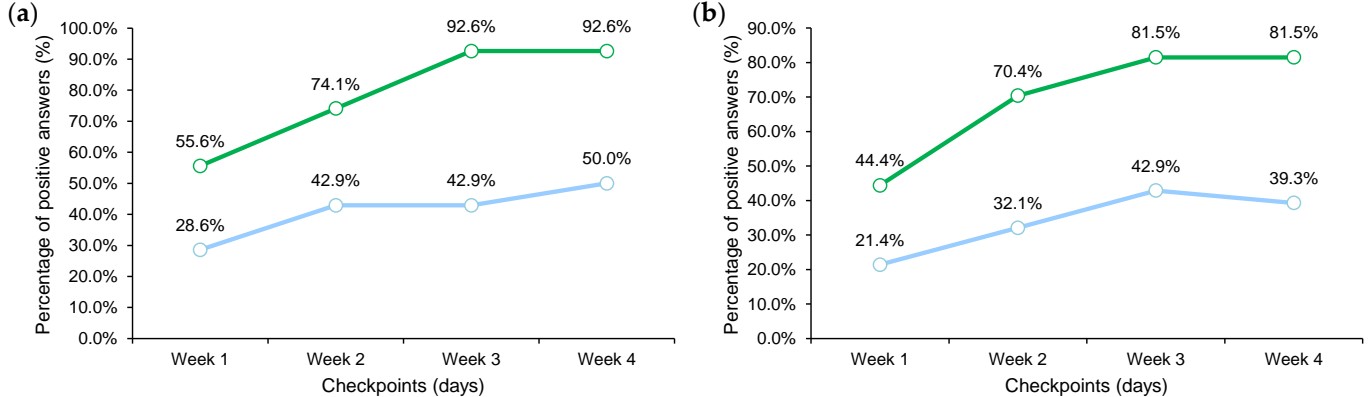

**Figure 4.** Weekly diary output. (**a**) Overall judgment of the treatment. (**b**) Overall judgment of the general skin conditions when compared to the skin conditions before treatment.

**Transepidermal water loss (TEWL).** TEWL was measured only on the adult population. Even if the TEWL variation was higher in the active treatment group (−11.6% vs. −4.4% at T14 and −17.2 vs. −8.1 at T28), the difference with the placebo product was not statistically significant. The obtained data indicate a trend of active treatment in improving the skin barrier. However, the small sample size ($n = 9$) does not allow for any further considerations to be made.

**Erythema index.** The intensity of skin redness (erythema index) was measured only in the adult population. The decrease in the erythema index was higher in the active treatment group (−10.9% vs. −6.1% at T14 and −20.2 vs. −12.0 at T28). The difference with the placebo product was statistically significant only after 14 days and not after 28 days of treatment. This trend can be justified by the relatively high variation of the individual response and by the small sample size ($n = 9$).

**Self-assessment questionnaire.** The output of the self-assessment questionnaire is reported in Table 4. For all the questionnaire items the active treatment was scored better than the placebo treatment.

**Table 4.** Overall (G1 + G2 + G3) self-assessment questionnaire after 28 day treatment with both the active and the placebo products. G1 newborns/infants/toddlers (6–36 months); G2 children (3–14 years); G3 adults (18+ years).

| No. | The Treatment Improved the Following Cutaneous Clinical Signs Related to AD: | Active | Placebo |
|---|---|---|---|
| 01a | Redness (body) | 85.2% | 71.4% |
| 01b | Redness (face) | 91.7% | 72.0% |
| 02a | Roughness (body) | 85.2% | 64.3% |
| 02b | Roughness (face) | 79.2% | 56.0% |

**Table 4.** *Cont.*

| No. | The Treatment Improved the Following Cutaneous Clinical Signs Related to AD: | Active | Placebo |
|---|---|---|---|
| 03a | Severe dryness (body) | 96.3% | 53.6% |
| 03b | Severe dryness (face) | 95.8% | 52.0% |
| 04a | Scratch marks (body) | 74.1% | 64.3% |
| 04b | Scratch marks (face) | 79.2% | 64.0% |
| **No.** | **The treatment improved the following symptoms related to AD:** | **Active** | **Placebo** |
| 05a | Incidents of pruritus/itching (body) | 88.9% | 71.4% |
| 05b | Incidents of pruritus/itching (face) | 95.8% | 48.0% |
| 06a | Intensity of pruritus/itching (body) | 85.2% | 57.1% |
| 06b | Intensity of pruritus/itching (face) | 79.2% | 52.0% |
| 07 | Sleeplessness | 95.0% | 60.9% |
| **No.** | **Other questions** | **Active** | **Placebo** |
| 08 | The treatment use overall improved the quality of my daily life | 92.6% | 53.6% |
| 09a | The treatment use reduced the appearance of skin irritation (body) | 77.8% | 50.0% |
| 09b | The treatment use reduced the appearance of skin irritation (face) | 79.2% | 44.0% |
| 10 | The treatment use reduced the flares incidences | 92.6% | 46.4% |
| 11a | The treatment use provided a soothing/relief of skin discomfort and tightness (body) | 92.6% | 57.1% |
| 11b | The treatment use provided a soothing/relief of skin discomfort and tightness (face) | 95.8% | 56.0% |
| 12a | After 4-week treatment, the skin is more supple, smooth and comfortable (body) | 92.6% | 50.0% |
| 12b | After 4-week treatment, the skin is more supple, smooth and comfortable (face) | 95.8% | 56.0% |

## 4. Discussion and Conclusions

The guidelines from the ETFAD recommend the use of mild detergents and the application of emollients. According to these guidelines, we studied the efficacy of a commercially available treatment composed of a cleanser, a face cream, and a body cream.

All the products contained ingredients targeting the epidermal barrier function as recommended for the treatment of mild-to-moderate AD [16].

Results from this study demonstrate the efficacy of the tested products in decreasing the overall AD severity through 28 days of treatment. Overall, the first results occurred within 14 days of treatment. The variation of the SCORAD index was statically significant, when compared to the placebo group, for the G1 group and overall group (where data from G1, G2, and G3 groups when pooled together). The responses to the active and placebo treatments were similar, even if the variation was smaller in the placebo group. This trend can be explained by the sufficient performance of the base (placebo) formulation and by the relatively small sample size.

Similar results were observed for the itching sensation (itching scoring during SCORAD) decrease in G2 and G3 groups. The overall (G2 + G3) itching sensation decreased by 46.6% and 70.1% after 14 and 28 days, respectively. The % variation was similar in both groups. Even if the percentage variation in the active group was higher at all the checkpoints when compared to the placebo group ($-22.4\%$ and $-33.1\%$, after 14 and 28 days, respectively) it was statistically significant only after 28 days of treatment.

In the weekly diary, the subjects reported a positive effect of the product in improving the itching sensation, skin redness, and skin dryness. All the parameters were decreased week by week through the four-week treatment period. The results of the overall itching sensation weekly scoring were consistent with the itching scoring during SCORAD. The decrease in the itching sensation between the active and the placebo products was statistically significant starting from week 3 of treatment.

The skin redness decrease was statistically higher in the active group after 4 weeks of treatment. A statistical difference between the active and the placebo groups was reported starting from the first week of treatment for skin dryness. In addition, the overall judgment of the treatment and the general skin condition was higher in the active group in a time-dependent manner (Figure 2). These results were confirmed also by the self-assessment questionnaire.

A trend toward the decrease was seen both for the transepidermal water loss and the erythema index. The results for these parameters are however not statistically or partially statistically significant due to the sample size.

Even though the trial was carried out on a small cohort of subjects, the data that resulted were sufficient to obtain statistically significant results. Another limitation was related to the placebo formulation choice. The chosen placebo formulation was very rich, resulting in minimal but statistically significant results.

In conclusion, the treatment was demonstrated to be effective in subjects with mild-to-moderate AD.

**Supplementary Materials:** The following supporting information can be downloaded at: https://www.mdpi.com/article/10.3390/cosmetics10040117/s1, Table S1: Inclusion and exclusion criteria; Table S2: Products qualitative formula.

**Author Contributions:** Conceptualization, V.Z. and V.N.; methodology, V.Z. and E.C.; software, V.N.; validation, V.N., M.P. and E.C.; formal analysis, M.P.; investigation, E.C.; resources, V.Z.; data curation, V.N.; writing—original draft preparation, V.N.; writing—review and editing, V.N. and E.C.; visualization, V.N.; supervision, V.N.; project administration, V.Z.; funding acquisition, V.Z. All authors have read and agreed to the published version of the manuscript.

**Funding:** This research was funded by Rontis Hellas S.A., Athens, Greece (now Dreavia AG, Zug, Switzerland).

**Institutional Review Board Statement:** The study was conducted in accordance with the Declaration of Helsinki and approved by the Ethics Committee of "Independent Ethical Committee for Non-Pharmacological Clinical trials" (protocol code 2021/17, approval date 23 December 2021).

**Informed Consent Statement:** Written informed consent has been obtained from the patient(s) to publish this paper.

**Data Availability Statement:** The data presented in this study are available upon request from the corresponding author. The data are not publicly available since they are the property of the sponsor of the study (Rontis Hellas S.A., Athens, Greece; now Dreavia AG, Zug, Switzerland).

**Acknowledgments:** The authors would like to express their gratitude to the Complife Italia staff, who contributed to the study and recruited the subjects, for their professionalism and support during the study development.

**Conflicts of Interest:** The authors declare no conflict of interest.

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
