# Peer review of "Efficacy of a Cosmetic Treatment in Decreasing the Mild-to-Moderate Atopic Dermatitis in Babies, Children, and Adults: A Pilot Study"

_cosmetics, doi:10.3390/cosmetics10040117_

Round 1
Reviewer 1 Report
The well-written paper is of interest to healthcare providers treating patients with atopic dermatitis, and the pilot study adds to the information on skincare for atopic dermatitis. The only suggestion I have to improve the manuscript I gave in my review: After defining the products, use generic names throughout the manuscript.
Reviewer 2 Report
In this article, Nobile et al analyzed the effect of a cosmetic treatment composed of a cleanser, and a body and face cream, in subjects (babies, children, and adults) suffering from mild-to-moderate AD. The product effect on AD clinical signs were investigated by scoring atopic dermatitis (SCORAD) index, subjective evaluation, skin erythema index and transepidermal water loss (TEWL) measurements. The products shown to be effective in improving the AD scoring by SCORAD in all the groups, and a trend towards the decrease of the erythema index and the TEWL in the adult population. An improvement of itching sensation, skin redness and skin dryness scoring was also reported by the subjects. Results from this study demonstrate the efficacy of the tested products in decreasing the overall AD severity through 28 days of treatment. Overall, the first results occurred within 14 days of treatment. In a randomized placebo-controlled trial, cosmetic treatments improved both objective and subjective measures of AD, which is considered an excellent result. There were few particular concerns. There are a few minor concerns as follows
minor concerns)
1) In line 30, you wrote "it is impaired." But no need for "it"?
2) In line 38, you wrote "SCOring Atopic Dermatitis (SCORAD) index." Would it be easier to capitalize "R" as in "SCORing"?
Not particular.
